# Immunosuppressive Agent Options for Primary Nephrotic Syndrome: A Review of Network Meta-Analyses and Cost-Effectiveness Analysis

**DOI:** 10.3390/medicina59030601

**Published:** 2023-03-17

**Authors:** Kei Nagai

**Affiliations:** University of Tsukuba Hospital Hitachi Social Cooperation Education Research Center, Hitachi 317-0077, Ibaraki, Japan; knagai@md.tsukuba.ac.jp; Tel.: +81-294-23-1111

**Keywords:** nephrotic syndrome, B cells, T cells, network meta-analysis, cost-effectiveness

## Abstract

Therapeutic options with immunosuppressive agents for glomerular diseases have widened with refinements to the Kidney Disease Improving Global Outcomes (KDIGO) guidelines from 2012 to 2021. However, international guidelines do not necessarily match the reality in each country. Expensive therapies such as rituximab and calcineurin inhibitors are sometimes inaccessible to patients with refractory nephrotic syndrome due to cost or regulations. Under the Japanese medical insurance system, rituximab is accessible but still limited to steroid-dependent patients who developed idiopathic nephrotic syndrome in childhood. Based on international KDIGO guidelines and other national guidelines, possible applications of immunosuppressive agents for nephrotic syndrome are comprehensively examined in this review. While rituximab has become the mainstay of immunosuppressive therapy for nephrotic syndrome, clinical trials have indicated that options such as cyclophosphamide, calcineurin inhibitors, and mycophenolate mofetil would be preferable. Given the rising number of patients with nephrotic syndrome worldwide, KDIGO guidelines mention the need for further consideration of cost-effectiveness. If the new option of rituximab is to be the first choice in combination with steroids for nephrotic syndrome, its cost-effectiveness should also be verified. Among the few studies examining the cost-effectiveness of treatments for nephrotic syndrome, administration of rituximab to young adults has been shown to be cost-beneficial, at least in Japan. However, further large-scale studies involving multiple facilities are needed to verify such findings. Network meta-analyses have concluded that the efficacy of rituximab remains controversial and confirmation through high-quality studies of large cohorts is needed. To this end, the mechanisms of action underlying immunosuppressive agents, both old and new, need to be understood and experience must be accumulated to evaluate possible effects and side effects.

## 1. Introduction

Nephrotic syndrome (NS) is frequently encountered and can show several variations in clinical course and pathological features. The disorder is further classified into primary NS, caused by primary glomerular disease, and secondary NS, caused by systemic disorders. While minimal-change NS (MCNS) is often incident in children, primary forms of NS in adults are categorized under three histological entities: idiopathic membranous nephropathy (MN); MCNS; and focal segmental glomerulosclerosis (FSGS) [1]. Secondary NS often necessitates treatment of the underlying disorder rather than specific treatment for the glomerular disease. Idiopathic NS is the most frequent glomerular disease of childhood [2], with an incidence of about 2–15 per 100,000 children younger than 16 years. The incidence of idiopathic NS in adults is relatively lower (0.6–1.2 cases per 100,000 adults). Along with the growth of the elderly population, the number of cases of glomerulonephritis may be increasing [3], incurring increasing economic burdens from the treatment for NS [4].

A vast body of experimental evidence, coupled with the clinical response of idiopathic NS to immunosuppressive therapy, clearly implicates the immune system in the disease pathogenesis of NS through the production of factors that damage the glomerular podocyte. Although the pathogenesis of MCNS, commonly seen in children, and the basis for immunosuppressive therapies remain unclear, studies suggest an imbalance of T helper 1 (Th1)/Th2 cytokines [5,6], Th17/regulatory T cell dysregulation [7,8], and abnormalities in B cells [9,10]. Autoreactive B cells and autoantibodies have also been shown to be involved in the development of NS, as clinical studies have identified anti-nephrin autoantibodies in a subset of adults and children with MCNS that align with results from animal studies [11,12], providing further support for an autoimmune etiology [13].

MN is the leading cause of NS in adults and an immune-mediated disorder with deposition of immunoglobulin (Ig)G and complement onto the glomerular capillary wall. A majority of idiopathic MN cases are currently defined by the presence of specific nephritogenic autoantibodies. A total of 70–80% of patients with MN show circulating autoantibodies to the phospholipase A_2_ receptor (PLA2R) [14,15], and 1–3% have circulating antibodies to thrombospondin type-1 domain-containing 7A (THSD7A) [16]. These discoveries support a pathological role of B cells as antibody producers [17,18]. In addition to directing antibodies against podocytes, B cells could also exert detrimental effects that indirectly alter the immunity of idiopathic NS patients by producing specific cytokines that can directly affect podocyte structure and functions or modulate T-cell homeostasis [10].

The availability of monoclonal antibodies against the B-cell surface antigen CD20, including rituximab (RTX), has enabled investigation into whether targeted B-cell depletion with inhibition of nephritogenic autoantibody production improves outcomes for patients with MN while avoiding the adverse effects of steroids and cytotoxic immunosuppressive agents. B-cell-targeting therapy with RTX is at least as effective as steroids and alkylating agents in achieving remission in MN, with available evidence suggesting that RTX is safer and better tolerated [19]. In other words, replacing traditional immunosuppressive regimens by specific, non-toxic modulation of B-cell immunity could lead to novel therapeutic paradigms for both pediatric NS and adult MN [19]. Evidence has emerged that levels of anti-podocyte antibodies (i.e., nephrin, PLA2R, THSD7A) parallel the induction of remission by B-cell depletion [13,20,21,22].

Guidelines differ in the recommendations or indications for therapeutic agents depending on when they were published and the country from which they were issued [1,23,24,25,26,27,28]. A previous descriptive NS classification, based largely on histological patterns, is increasingly being replaced on the basis of new pathogenic insights [3]. This change in classification has led to the development of more and more cause-driven treatments. Not only the discovery of autoantibodies in MN, but also the recent involvement of autoantibodies in MCNS has led to a shift toward pathophysiology-based therapy. However, determining the superiority of new treatment regimens over old regimens is not easy. Of course, using drugs that have no therapeutic effect on NS is clearly inappropriate, but when it comes to drugs that do have certain therapeutic effects, having various treatment options is beneficial in terms of possible future drug manufacturing, distribution, and adaptability to low-income countries.

This review article provides an overview of the basic mechanisms underlying immunosuppressive agents in the treatment of idiopathic NS, both old and new, and discusses the advantages, limitations, and comparative efficacies of immunosuppressive agents by reviewing results using a new analytical method: the network meta-analysis. The aim is to promote consideration of preferable treatments in terms of cost-effectiveness among immunosuppressive agents, with a particular focus on RTX.

## 2. Current Applications of Immunosuppressive Agents for Primary NS Listed in Guidelines

First, the content of this section is a summary of guidelines for NS written in the English language and identifiable from a search of PubMed. Recommendations for immunosuppressive agents show differences according to publication time and country of origin. Although drugs traditionally used for NS, such as herbal medicines and anti-rheumatic drugs such as leflunomide, may be considered, such agents are not described herein based on the scope of the review. Similarly, alternative B-cell-targeting CD20 therapies such as ofatumumab, obinutuzumab, and ublituximab were considered [29]. The monoclonal antibody belimumab specifically targets the soluble form of B-lymphocyte stimulator (BlyS, also known as BAFF), a tumor necrosis factor superfamily ligand that plays a critical role in the differentiation and homeostasis of B lymphocytes [30,31]. Early-stage MN may be predominantly mediated by CD20+ cells and thus susceptible to targeted therapy, whereas more advanced disease is mediated by autoreactive memory plasma cells resistant to anti-CD20 monoclonal antibodies but possibly sensitive to proteasome inhibitors targeting plasma cells, such as bortezomib [32]. However, recommendations of these alternative options are not well described in guidelines.

There is no question as to the importance of immunosuppression to reduce the use of steroids, not only for NS but also for many other inflammatory diseases and immunological diseases, because of the need for long-term administration of steroids. Corticosteroids (most commonly prednisone) remain the cornerstone of treatment for primary NS. While any type of NS can be treated by corticosteroid, the risk of side effects from long-term use are a concern. In children, the optimal treatment strategy therefore aims at employing the lowest cumulative doses of corticosteroid and the safest and most effective corticosteroid-sparing agents to maintain remission, as children may be vulnerable to the side effects of high cumulative doses of steroid [33]. Young patients are at risk of developing obesity [34], growth impairment [35], behavioral alterations and attention problems [36], as well as reduced quality of life. Immunosuppressive agents are now often used in combination with corticosteroids rather than as monotherapies for inducing remission of primary NS. Table 1 presents a list of immunosuppressive agents for NS other than corticosteroids.

As most pediatric NS is treated without renal biopsy and considered to represent MCNS, the pathological type of NS is not always diagnosed before treatment. Therapeutic algorithms have therefore been described for steroid-sensitive or -resistant NS [28,37]. The initial treatment of pediatric NS is prednisolone based on the guidelines. When there is insufficient response to corticosteroids, a kidney biopsy is performed before adding alternative immunosuppressive agent. NS in adults is encouraged to be diagnosed by serological study or histological examination. Both guidelines from the Japanese Society of Nephrology and KDIGO separate disease types, including MCNS, MN, and FSGS. Despite of differences in therapeutic algorithms, cyclophosphamide (CPA), cyclosporine (CsA), tacrolimus (TAC), and mycophenolate mofetil (MMF) for NS in children and MCNS in adults were used throughout the guidelines (Table 1). On the other hand, a drug option has been removed over time. Chlorambucil (CLB) is a similar drug class to CPA but is inferior to CPA because of side effects of infections and malignancies [24,26,38], although the evidence supporting CPA over CLB is not strong. CLB has not been approved in certain countries including Japan, due to side effects such as the development of hematological malignancies.

In the current pediatric nephrotic treatment setting, RTX is given after kidney biopsy to patients with resistant NS who do not respond to second-line drugs such as cyclosporine. Particularly in KDIGO 2012, RTX was not listed as a first-line therapy because of the lack of evidence from randomized controlled trials (RCTs). This rapid paradigm shift in the use of RTX for NS, as emphasized in KDIGO guidelines from 2012 to 2021, raises two questions [1,26]. The first is whether the positioning of traditionally used immunosuppressive drugs should be changed as new treatments are recommended, and the second question is whether B-cell/autoantibody-targeted therapies, as the centerpiece of the paradigm shift, can be expected to have therapeutic effects commensurate with the generally high cost. While renal healthcare for NS continues to evolve and research findings are being shared around the world, equal access to treatment is not guaranteed [39]. International guidelines from KDIGO have thus consistently been thoughtfully written with regional differences in economic characteristics in mind [1,26].

This review therefore discusses the characteristics of regions. Use of levamisole (LEV) is unique to NS in children living in certain countries, including India [40,41], as broadly described in guidelines [1,25,26,27,28]. That RCTs are needed to clarify the role of LEV in frequently relapsing/steroid-dependent MCNS in adults was mentioned in KDIGO 2012 through to KDIGO 2021. Very few well-designed comparative studies of LEV for NS in adults have been reported, but the proposed molecular mechanisms appear to be supported by an experiment using puromycin-aminonucleoside-treated cells [42,43]. For Japanese children, three immunosuppressive agents commonly used in the treatment of frequently relapsing and steroid-dependent NS are CsA, CPA, and MZB, instead of the MMF listed in international guidelines [23,26,28]. In adults, apart from in MCNS and idiopathic MN, use of RTX in FSGS is not recommended in KDIGO guidelines, while Japan allows the use of RTX in steroid-dependent frequently relapsing NS, including FSGS, if the condition of childhood onset is met [24].

This comparative summary is not able to cover the guidelines of all countries but does indicate that individual national guidelines will not always be identical to the current recommendations in international KDIGO guidelines.

## 3. Options for Immunosuppressive Agents

Immunosuppressive agents used in therapy for NS regulate lymphocyte number and function. B-cell-targeting therapies have been the focus of many recent efforts, following the discovery of pathogenic antibodies [10], but T cells and B cells can interact with each other [44]. B-cell depletion can indirectly affect T-cell subsets interacting with B cells in lymphoid follicles [45]. B-cell regulation may affect glomerular podocytes, which may similarly regulate T-cell function [46,47]. On the other hand, CsA, which regulates T-cell signaling, is also known to have the potential to indirectly decrease B-cell function [48,49]. This review does not intend to mention further such indirect mechanisms in detail, but it is necessary to show that even one direct point of action, such as a single molecular target of an immunosuppressive agent, has the possibility of affecting the broad mechanisms of the immune system. However, this section concisely delineates how each immunosuppressive agent affects pathogenic players in primary NS to clarify options for any type of NS, regardless of the results of designed RCTs.

The treatment of NS with a wide range of immunosuppressive agents has been attempted for a long time, leading to a growing body of knowledge, as outlined below. As with virtually every one of the immunosuppressive agents used in the treatment of NS, the risks of treatment must be weighed against potential benefits on the basis of drug exposures and individual patient factors, including age, cancer, immunological complications, and comorbid conditions such as obesity and diabetes. The availability of specific agents will also vary between countries and regions. To present the large number of drugs available as fairly as possible, the following passages only include very brief information, actual uses, and regional availability.

### 3.1. Cyclophosphamide (CPA)

CPA is a cytotoxic agent exerting strong actions through the alkylation of purine bases, representing the most common drug used in the treatment of NS. The damage to DNA synthesis induces apoptosis or altered function in both B and T cells [50]. CPA has historically been used to prolong remission in MCNS, FSGS, and MN [51,52]. Major adverse effects include infertility, malignancies, and bone marrow suppression, and use of this drug requires careful monitoring of blood counts and adjustment of doses relative to the degree of renal impairment. CPA is currently one of the most commonly used anticancer drugs in the world, indicated for various types of cancer, and is available in almost all regions.

### 3.2. Chlorambucil (CLB)

CLB is another alkylating agent, although less commonly used than CPA because of some major differences in the frequency of adverse events. As for CPA, CLB can induce apoptosis or alter the function of B and T cells. This agent has been used in the treatment of patients with MN, and the original regimen was developed by Ponticelli et al. [53]. The drug is not approved for use against NS in various countries, including Japan, where it is only available as an anticancer drug for veterinary use.

### 3.3. Levamisole (LEV)

LEV is an anthelmintic agent that is known to have immunomodulatory properties that enhance humoral immune responses and polymorphonuclear cell survival. On T cells, LEV has indirect effects via activation of antigen-presenting cells, eliciting CD4^+^ T-cell proliferation and Th1 responses through the production of interleukin-18 and interferon-γ [54]. LEV has thus been used for treating tumors and various immunological disorders such as Behçet’s syndrome and NS over the past three decades. Recently, Sinha et al. [55] presented results from an RCT comparing MMF and LEV administered for 12 months to Indian children with NS. Both agents showed similar efficacy in reducing the frequency of relapses, maintaining remission, and successfully sparing patients from corticosteroid use [54]. The low cost of LEV makes this agent a useful option, particularly in low-income and low-resource settings. However, LEV is unavailable in some countries; the drug was withdrawn from the United States market in 1999 because of adverse effects relating to agranulocytes and anti-neutrophil cytoplasmic antibody (ANCA)-positive small vessel vasculitis [56,57], and from the European market in 2004 because of a lack of clear indications [54]. In Japan, the drug has not been approved for use against NS but is available as an anthelminthic drug for veterinary use. Nevertheless, the lack of nephrotoxicity is considered a major advantage for its use in NS.

### 3.4. Rituximab (RTX)

RTX is a genetically engineered chimeric murine/human monoclonal antibody directed against CD20, an antigen found on the surface of normal pre-B cells and mature B cells. Therapy with this agent results in highly effective depletion of CD20-positive B cells via inhibition of cell proliferation and direct induction of B-cell apoptosis [58]. RTX is standard treatment for children with frequently relapsing or steroid-dependent NS, particularly when other immunosuppressive drugs have failed to achieve long-standing full remission [59]. Coupled with emerging generic products, the market for RTX is expanding worldwide. Unlike other immunosuppressive drugs discussed in this section, since no oral form of RTX is available, this agent is administered as outpatient chemotherapy by intravenous infusion or as inpatient treatment for monitoring allergic reactions. To ensure the efficacy of B-cell depletion, numbers of CD20- or CD19-positive cells need to be periodically checked by flow cytometry.

### 3.5. Calcineurin Inhibitors (CNIs): Cyclosporine (CsA) and Tacrolimus (TAC)

CsA and TAC are CNIs that suppress the immune response by down-regulating T-cell activation. These agents specifically block calcium-dependent T-cell receptor signaling transduction, thereby inhibiting the transcription of IL-2 as well as other pro-inflammatory cytokines, in both T cells and antigen-presenting cells [60]. A direct effect of CNIs on the actin cytoskeleton of podocytes has been also suggested [61]. Therapies with TAC and CsA show similar efficacy for NS [62]. Since the discovery of TAC as an immunosuppressive agent, much better organ transplant survival rates have been achieved, along with expanded indications for many immune diseases including psoriasis, aplastic anemia, myasthenia gravis, and atopic dermatitis. This drug has become available worldwide. Because major adverse effects include renal impairment, diabetes mellitus, and hypertension, caution is warranted in the case of long-term or high-dose administration.

### 3.6. Mycophenolate Mofetil (MMF)

MMF is a relatively new immunosuppressive agent that is already available worldwide. Similar to azathioprine (AZP), MMF is a reversible inhibitor of inosine monophosphate dehydrogenase, a critical enzyme involved in de novo purine synthesis that is required for lymphocyte division [63]. The selectivity of MMF for inhibiting B- and T-cell proliferation underlies the reduced toxicity of MMF compared with alkylating agents that affect all dividing cells. MMF is currently considered by many pediatric nephrologists as the “standard of care” for steroid-sparing treatment in children with primary NS because of its universal availability, lack of nephrotoxicity, and well-documented safety profile [64].

### 3.7. Mizoribine (MZB)

MZB is an imidazole nucleotide that exerts selective inhibitory effects on inosine-5-monophosphate dehydrogenase, an enzyme in the de novo purine nucleoside synthesis pathway [65,66]. These effects are very similar to those of MMF and result in suppression of T- and B-cell proliferation. Moreover, a recent study suggested that MZB directly prevents podocyte injury and preserves nephrin structure, leading to a reduction in urinary protein [67]. In Japan, MZB is approved under insurance for steroid-resistant NS, in place of MMF. This agent is known not to cause serious adverse events due to its less myelosuppressive, less hepatotoxic effects [68] and is considered beneficial when administered over a long period [69].

### 3.8. AZP

The mechanism of action of AZP involves its degradation to 6-mercaptopurine (6-MP) in the body; this purine antagonist non-specifically inhibits leukocyte proliferation by interfering with nucleotide synthesis [70]. As an imidazole derivative, AZP may also be involved in the reduction of nuclear factor of activated T cells (NFAT) signaling following T-cell receptor activation [71]. In addition to the effects of AZP on purine nucleotide synthesis, AZP can directly promote apoptosis and inhibit the proliferation pathway of lymphocytes in vivo and in vitro [72]. AZP is a less toxic immunosuppressive agent than cytotoxic drugs such as CPA, CLB, and CNIs, and has been used in the treatment of patients with idiopathic MN in combination with corticosteroids, although the magnitude of effect is not considered high [73].

## 4. Emerging Network Meta-Analysis of Efficacy of Immunosuppressive Agents for NS

Recommendations from any guidelines are basically derived from the results of RCTs, which are designed to allow drug-to-drug or drug-to-placebo comparisons to clarify the independent effects of the drug of interest. Particularly in fields where a large number of therapeutic agents are used, such as for primary NS, clinical trials that directly compare one therapeutic agent to another are limited in terms of the number of trials and cases, and the results of network meta-analysis can be helpful in such cases.

Network meta-analysis, like conventional meta-analysis, is an analytical method that statistically integrates the results of multiple clinical studies. Whereas conventional meta-analysis is limited to comparisons of two parties, network meta-analysis can compare three or more parties [74]. The results of recent network meta-analyses have been published, although the number of studies focusing on NS in children [73,75,76] and idiopathic MN [77,78,79,80,81,82] remains small (Table 2). Most selected studies in the network meta-analyses comprise groups on corticosteroid monotherapy or combined therapy with immunosuppressive agents. Use of corticosteroid is not counted in Table 2, in accordance with the concept of this review.

The number of RCTs incorporated varies widely, depending on the age of the patients, type of NS, drugs of interest, language and year of publication, and other factors (Table 2). In particular, the largest study, by Dai et al., included Chinese-language literature [80]. Because meta-analyses by nature employ older studies, the overwhelming majority of immunosuppressive agents in these analyses have been CPA, followed by CNIs, and very few RTX.

In children, three analyses were published [75,76]. Fu et al. analyzed four commonly used immunosuppressive agents (CPA, CsA, TAC, and MMF) from 7 RCTs for refractory pediatric NS [75]. They revealed that treatment with MMF had the greatest odds of relapse compared with TAC (pooled odds ratio (OR): 49.2), CPA (OR: 72.1), and CsA (OR: 11.4). Rank probability analysis found CPA was the best treatment with the lowest relapse rate compared with other treatments (rank probability: 0.58), and TAC was ranked second best (rank probability: 0.38). Li et al. examined 18 RCTs and identified CsA, TAC, RTX-CsA dual therapy, and MMF as efficacious treatments for achieving complete remission [73]. RTX has not yet been suitably evaluated because only one study was included, in which RTX was used as a concomitant drug. In 2019, Tan et al. finally examined 3 trials on RTX from 26 selected RCTs in a network meta-analysis. At 6 months, treatment with CPA, CLB, LEV, or RTX offered better efficacy than placebo/non-treatment groups (ORs: 0.09, 0.03, 0.28, and 0.07, respectively), indicating that CPA, CLB, and RTX may be preferable for use in children with NS [76]. Nevertheless, the authors concluded that additional evidence regarding the safety and efficacy of RTX in children with frequently relapsing/steroid-dependent NS is needed [76].

In adults, the scope for most studies is idiopathic MN, and six network meta-analyses were identified [77,78,79,80,81,82]. Depending on study design and agents of interest, the number of RCTs has varied, and the drugs considered have broadened to include leflunomide. Consistently, CPA followed by CNIs were the most reliable for remitting idiopathic MN in adults. Zheng et al. [79] first included a trial on RTX in a network meta-analysis and compared 13 treatment regimens including the seven immunosuppressive agents (CPA, CLB, RTX, CsA, TAC, MMF, and MZB). The results showed that most regimens other than MZB showed significantly higher probabilities of remission when compared with non-immunosuppressive therapies (control group), with risk ratios (RRs) of 1.9 for CPA, 1.7 for CLB, 1.8 for RTX, 2.0 for CsA, 2.0 for TAC, and 1.9 for MMF, respectively. This result indicates that RTX may be acceptable, but CPA and TAC are superior to other immunosuppressive agents. However, RTX was merely superior to the control group in remission, without evident advantages over other options. Liu et al. [81] studied comparisons among 12 treatment regimens, with 51 RCTs including 5 trials on RTX. Compared with the control group, most regimens demonstrated better effects in remission, with RRs of 2.2 for CPA, 1.9 for RTX, 1.6 for CsA, and 2.1 for TAC, respectively [81]. However, the authors were also concerned that compared with other immunosuppressive agents, RTX was more expensive, hindering its popularization in economically underdeveloped countries to a certain extent [81].

In terms of epidemiology, mean onset age of primary glomerular disease has appeared to be on the rise, the proportion of NS has increased, and the most common pathological types have altered from IgA nephropathy to MN, based on biopsy studies [83,84,85]. Moreover, diagnosis has recently been made using a combination of antibody tests (anti-PLA2R or anti-THSD7A antibodies) and kidney biopsy. This indicates that the antibody-positive idiopathic MN patients can be added without renal biopsy. Therefore, concerns have been raised that many more cases of primary MN will be encountered worldwide. As seen in the case of MN, the possible use of any type of RTX for NS, even if not currently applicable (e.g., adult-onset NS in Japan, FSGS and other steroid-refractory NS), is expected to increase in the future. The growing application of RTX will further increase the accuracy of discussions regarding the superiority of RTX compared to classical and immunosuppressive agents through network meta-analysis.

## 5. Cost-Effectiveness Studies in Treatment of NS

Immunosuppressive agents differ not only in their effect against NS and adverse effects, but also drug cost. These factors can be indicators for recommendations for therapy against NS in terms of medical costs incurred for various tests, hospitalizations, and emergency room visits related to recurrence of NS, and side effects of immunosuppressive drugs. The issue of healthcare costs straining the economy is a global concern and is mentioned in the chapters of the KDIGO guidelines as a constant consideration [1,26]. As stated, the guidelines target a broad audience of clinicians treating glomerular disease in all parts of the world while being mindful of policy and cost implications. Most of the drugs recommended are available at low cost in many parts of the world. These include corticosteroid, AZP, and CPA tablets. Monitoring (e.g., by regularly checking blood counts) is also cheap and widely available [26]. The cost of some agents (e.g., CNIs, MMF, and RTX) remains high, but the development and marketing of generic and biosimilar agents is now rapidly reducing costs. However, care must be taken to ensure that variations in bioavailability with these less expensive generic agents do not compromise effectiveness or safety.

Currently, debate on the financial burdens of healthcare for NS cannot be considered “active” per se. However, in clinical practice, examples of strict indications for RTX use, as well as strict scrutiny of drug prices, can be presented. The National Health Service (NHS) England had not recommended RTX for use in adult MN, but carefully reviewed the evidence and allowed the treatment of idiopathic MN with RTX [86,87,88]. In October 2022, they concluded that the evidence available at that time was sufficient to allow the treatment, but the criteria were strictly determined. Instead of being uniformly applicable, primary immunosuppressive therapy with RTX can be considered for patients who meet all the following inclusion criteria: (1) Diagnosis of IMN. Diagnosis is made using a combination of antibody tests (anti-PLA2R or anti-THSD7A antibodies) and kidney biopsy. (2) Contraindications or intolerances to cytotoxic or steroid therapy, or an inability to comply with monitoring requirements for cytotoxic therapy. (3) An estimated glomerular filtration rate > 20 mL/min/1.73 m^2^. (4) Agreement within the multi-disciplinary team that RTX at the specified frequency and dose represents the most appropriate treatment option. Importantly, NHS England also reserves the right to review policies where the supplier of an intervention is no longer willing to supply the treatment to the NHS at or below this price and to revise policies where the supplier is unable or unwilling to match price reductions in alternative therapies.

Nevertheless, some studies have undertaken economic evaluations of pharmacotherapy for NS. Among those, one RCT tested the induction of remission with steroid monotherapy. The object of that analysis by Webb et al. using data from the prednisolone in nephrotic syndrome (PREDNOS) trial [89] was to determine whether an initial 16-week extended course of prednisolone treatment increased the time to first relapse in children with steroid-sensitive NS compared with the 8-week standard course; a cost-effectiveness analysis was also performed by comparing costs and quality-adjusted life years (QALY) for the two regimens. Extended course was associated both with a mean increase in quality of life (0.016 QALY, 95% confidence interval (CI) −0.005 to +0.037) and a cost savings (difference GBP −1673; 95%CI GBP −3455 to GBP +109), while clinical outcomes were unimproved with an extended course. They concluded that the extended course of prednisolone could offer a cost-effective use of healthcare resources. However, critical concerns about adverse events from extended prednisolone treatment were not addressed [89].

Simon explored actual economic burdens comprising medical and non-medical costs of NS for patients and family caregivers in the United States. The cohort comprised adults with NS and family caregivers of children with NS for at least one year [90]. This survey was the first to characterize out-of-pocket expenses in the United States and revealed median annual medical costs of USD 3464 (interquartile range, USD 844–5865) for adult patients and USD 1687 (interquartile range, USD 1035–4763) for caregivers. The bulk of financial and time costs was for procurement and preparation of meals for NS. The study concluded that adults and caregivers of children with NS face substantial disease-related direct and indirect costs beyond those covered by insurance.

Thus, other than immunosuppressive drugs, cost-effectiveness considerations are important for renal healthcare, including use of corticosteroid use and diet. This review further focuses on reports involving immunosuppressive agents (Table 3). Long-term cost of treatment is another factor to consider when selecting a steroid-sparing agent with relatively high cost. Although population growth has not yet been shown to correlate precisely with increases in the number of patients with NS, the population, but not prevalence, of any non-communicable disease such as primary glomerular disease may increase with a population explosion [91,92]. If treatment for NS according to a guideline is expensive, coupled with population growth and widening economic disparities, equitable supply to patients will become difficult. This may be particularly relevant in countries with large populations such as India [93,94], where public health coverage is incomplete and the burden of drug costs is almost entirely borne by the usually low-income families [95]. Since the emergence of RTX for NS, analyses of cost-effectiveness have gained prominence [96,97]. Recently, Takura et al. evaluated the cost-effectiveness of treating NS with RTX and suggested that this therapy may be superior to conventional treatment from a health economics perspective, at least in Japan [96] (Table 3).

For instance, in a case of MN, treatment with immunosuppressive agents is associated with high cost, including therapy, monitoring, and management of side effects. Conversely, induction of renal replacement therapy, recurrence of NS, and hospitalization are possibly associated with much higher costs and even more adverse effects than immunosuppressive agents. The current goal of analyses of cost-effectiveness at this time is thus to clarify whether RTX is worth the cost as a treatment with a high remission rate and low rate of progression to end-stage renal failure. Pre-post cost effectiveness analyses are used to compare the economic efficiency of medical interventions before and after treatment [99]. When costs are low and effectiveness is high for the replacement technology, the pre-post cost effectiveness is said to be “dominant” [99].

Takura et al. investigated the cost-effectiveness of RTX for adult NS [96]. In 30 patients with mean age of 29.1 years, before (with corticosteroid and immunosuppressive agents) and after introducing RTX therapy, a significant improvement in relapse rate was seen, from a mean of 4.30 events before administration to a mean of 0.27 events after administration for 24 months, indicating significantly better renal prognosis with RTX. As a result, total monthly medical costs decreased from USD 2923 to USD 1280, and the pre-post cost effectiveness was “dominant”. They concluded that switching from the previous pharmacotherapy to RTX effectively improved the relapse rate and was an improvement from an economic perspective as well. The analysis was based on a regimen followed at a single institute and a cohort with young adult subjects, so the actual state of treatment nationwide was not fully reflected. Further large-scale studies involving multiple facilities are needed to verify the findings [96]. Hamilton et al. developed a decision-analytic model to estimate the cost-effectiveness of RTX therapy versus the standard of care for idiopathic MN in middle-aged patients, in the form of conventional methylprednisolone and CPA, recognized as a modified Ponticelli regimen [97]. At 1 year post-treatment, RTX therapy “dominated” the modified Ponticelli regimen (mPR) by +0.002 incremental QALY and GBP −748 incremental cost. Moreover, at 5 years post-treatment, RTX was still cheaper than mPR (GBP −1355) but with a loss of QALYs (−0.014 QALYs), giving an incremental cost-effectiveness ratio (ICER) of GBP +95,494, reflecting better cost-effectiveness from treatment. That result indicated that RTX has potential as a cost-effective treatment over the short and medium terms despite the high single-dose cost, although this analysis was based on modeling.

Ramachandran et al. investigated steroid-dependent or -resistant MCNS and FSGS patients with CNI dependence. That dose-testing clinical study aimed to minimize RTX use in a total of 24 patients followed-up for 12 months. They received a standard dose (375 mg/m^2^) at entry, then CD-19 levels were carefully monitored monthly, and patients with CD19 levels > 5/μL and/or >1% received an additional low dose (100 mg) of RTX [98], based on the philosophy and experiences that high-dose RTX administration may not be as necessary as in lymphoma treatment [98,100,101]. Although the observation was neither controlled nor examined based on cost-effectiveness analyses, the results were very informative regarding reducing the cost of maintenance for NS by RTX. However, it is hard to conclude that RTX is cost-effective, given the different dosing regimens in each study and the fact that drug prices vary by country and time of analysis.

Most recently, Dai et al. compared efficacy and costs with more than two arms of treatment for the first time [80]. They undertook a cost-effectiveness analysis using the decision tree model. This model was constructed to simulate the remission and recurrence of idiopathic MN in the same population after different treatments (CPA, CLB, RTX, TAC, CsA, MMF, or leflunomide) [80]. The length of the study period was set to six months and the initial state was assumed to be the disease state. The cost-effectiveness ratio (CER) results in China showed that CPA was the lowest (USD 42 per unit utility), followed by leflunomide (USD 49), and MMF (USD 830). Conversely, RTX exhibited the highest CER (USD 3042). According to CER analysis, immunosuppressive agents other than CTX, leflunomide, and TAC were not recommended because of the high costs without effects exceeding the expected value. In particular, the effects of RTX for idiopathic MN appear promising, but the failure to find any significant result in this study may due to insufficient sample size, or these agents may not actually be effective in achieving remission of NS for certain populations. The investigators therefore concluded that the efficacy of RTX needs to be confirmed with more high-quality studies of larger cohorts.

## 6. Conclusions

Many immunosuppressive agents, new and old, are used in the treatment of primary NS. Guidelines have recommended several agents at different times and in different regions. With the increasing understanding of the pathophysiology of NS, particularly in MCNS and MN, a shift toward specific therapeutic targets that are less prone to side effects is underway. Representative of this shift are B-cell targeted therapies, including RTX, as indicated by the transition in the KDIGO guidelines from 2012 to 2021. A network meta-analysis summarizing RCTs of numerous immunosuppressive agents shows that the number of trials has not reached the point where RTX is superior, and the efficacy of CPA and CNIs is still reliable. Moreover, a new and expensive therapeutic agent may be difficult to obtain or may not covered by insurance in some regions, so the international KDIGO guidelines always present options that allow for differing regional characteristics. Recent cost-effectiveness analyses are showing the superiority of RTX, and further analyses are needed to predict future changes in disease prevalence and drug price. Continued accumulation of the human experience in treating NS is important, along with selection of available immunosuppressive agents based on knowledge of the principles of immunological efficacy.

## Figures and Tables

**Table 1 medicina-59-00601-t001:** List of immunosuppressive agents other than corticosteroid (CS) in recent guidelines for nephrotic syndrome (NS) in children and adults.

Guideline	Children	Adults
KDIGO 2012 [1]	CPA, CLB, LEV, CsA, TAC, MMF (RTX not recommended as treatment option for SRNS, due to lack of RCTs; MZB and AZP considered ineffective)	MCNS: CPA, CsA, TAC, MMF
		MN: CPA, CLB, CsA, TAC, MMF
		FSGS: CsA, TAC
		(RTX not recommended as treatment option for SRNS, due to lack of RCTs)
JSPN 2013 [23]	CPA, RTX, CsA, TAC, MMF, MZB	-
JSN 2020 [24]	-	MCNS: CPA, RTX, CsA, TAC, MMF, MZB, AZP
		MN: CPA, CLB, RTX, CsA, TAC, MMF, MZB, AZP
		FSGS: CPA, RTX, CsA, MMF, MZB, AZP
		(CLB not approved; TAC not covered by insurance; RTX only for child-onset cases)
GSPN 2020 [25]	CPA, LEV, RTX, CsA, TAC, MMF	-
KDIGO 2021 [26]	CPA, LEV, RTX, CsA, TAC, MMF (CLB inferior to CPA)	MCNS: CPA, RTX, CsA, TAC, MMF
		MN: CPA, RTX, CsA, TAC (MMF not discussed because of insufficient RCTs; MZB considered only for maintenance)
		FSGS: CsA, TAC
ISPN 2021 [27]	CPA, LEV, RTX, CsA, TAC, MMF	-
IPNA 2022 [28]	CPA, LEV, RTX, CsA, TAC, MMF	-

Abbreviations: KDIGO, Kidney Disease Improving Global Outcomes; JSPN, Japanese Society for Pediatric Nephrology; JSN, Japanese Society of Nephrology; GSPN, German Society for Pediatric Nephrology; ISPN, Indian Society of Pediatric Nephrology; IPNA, International Pediatric Nephrology Association; CPA, cyclophosphamide; CLB, chlorambucil; LEV, levamisole; RTX, rituximab; CsA, cyclosporine; TAC, tacrolimus; MMF, mycophenolate mofetil; MZB, mizoribine; AZP, azathioprine; SRNS, steroid-resistant NS; RCTs, randomized controlled trials; MCNS, minimal-change NS; MN, membranous nephropathy; FSGS, focal segmental glomerulosclerosis.

**Table 2 medicina-59-00601-t002:** List of network meta-analyses for NS regarding use of immunosuppressive agents mostly combined with CS.

Author Name, Year, Disease	Total No. of Studies	IS Agents of Interest (No. of Studies)	Highlights
Fu, 2017, child NS [75]	7	CPA (2), CsA (5), TAC (3), MMF (3)	CPA and TAC are the best and second-best agents for relapsing NS.
Ren, 2017, idiopathic MN [77]	36	CPA (15), CLB (9), CsA (6), TAC (5), MMF (4), MZB (2), AZA (3), leflunomide (1)	CPA and CLB reduce risk of renal failure. CsA and TAC increase rate of proteinuria remission.
Li, 2017, child NS [73]	18	CPA (8), CLB (1), RTX (1), CsA (8), TAC (4), MMF (3), AZA (1), leflunomide (1)	TAC and CsA may be preferred initial treatments for children NS. MMF may be another option.
Jiang, 2018, adult NS [78]	7	CPA (3), CsA (4), TAC (2)	TAC and CsA are the best and second-best agents for inducing remission.
Tan, 2019, child NS [76]	26	CPA (8), CLB (4), LEV (10), RTX (3), CsA (4), MMF (3), AZA (2), vincristine (1)	CPA may be preferred initially in children with NS. CLB and RTX may be acceptable.
Zheng, 2019, idiopathic MN [79]	48	CPA (23), CLB (11), RTX (1), CsA (10), TAC (16), MMF (6), MZB (1), leflunomide (2)	TAC and CPA are superior to other IS agents.
Dai, 2021, idiopathic MN [80]	75 (24 in English; 51 in Chinese)	CPA (57), CLB (5), RTX (3), CsA (16), TAC (30), MMF (15), leflunomide (11)	Remission rate highest with TAC despite high single-dose cost.
Liu, 2022, idiopathic MN [81]	51	CPA (28), CLB (7), RTX (5), CsA (14), TAC (13), MMF (8), MZB (1), AZA (1), leflunomide (4)	TAC + MMF performed best in remission. Little advantage from MZB, AZA, or leflunomide in treating patients with idiopathic MN compared with control.
Bose, 2022, idiopathic MN [82]	56	CPA (26), CLB (10), RTX (4), CsA (12), MMF (7), MZB (3), AZA (3), leflunomide (1),	Comparative effectiveness and safety of IS compared to CPA are uncertain in adults with idiopathic MN.

The order of listing for reagent names does not indicate priority, strength of recommendations, or effectiveness. Most selected studies in network meta-analyses comprised groups with CS monotherapy or combined therapy with immunosuppressants. Use of CS is not counted in this table, according to the concept of this review. Abbreviations: IS, immunosuppressive.

**Table 3 medicina-59-00601-t003:** Studies related to cost-effectiveness in treatment with IS agents for primary NS.

First Author, Nation, Year Disease	Intervention/Exposure, Study Design/Model	Result	Interpretation
Takura, Japan, 2017 Adult NS [96]	Before and after RTX Observational cohort	Improvement in relapse rate from 4.3 to 0.3 events after administration of RTX, and total medical costs decreased from USD 2923 to USD 1280 per month.	Treatment with RTX is possibly superior to previous pharmacological treatments from a health economics perspective.
Hamilton, UK, 2018 Idiopathic MN [97]	Decision-analytic model RTX Modified Ponticelli regimen (CPA + CS)	At 1 year post-treatment, rituximab therapy dominates CPA + CS. At ≥5 years post-treatment, RTX is cheaper than CPA + CS.	RTX is not more expensive than gold-standard treatment and is cheaper.
Ramachandran, India, 2019 Adult SDNS/SRNS with CNI dependence [98]	Single-arm cohort CD19 cell monitored RTX regimen: 100 mg additional low dose if necessary after single dose of 375 mg/m^2^.	Average cost of monitored RTX regimen in the first year is USD 487. Conventional regimen (375 mg/m^2^ × 4 doses) is USD 1415. Over three-quarters of patients maintain remission.	CD19 monitored RTX regimen seems safe and cost-effective for remission maintenance in adult NS with CNI dependence.
Dai, China, 2021 Idiopathic MN [80]	Decision tree model to simulate remission and recurrence after each IS agent: CPA, CLB, RTX, CsA, TAC, MMF, leflunomide	CER in China showed CPA was cheapest (USD 42 per unit utility), followed by leflunomide (USD 49) and MMF (USD 830). RTX exhibited the highest CER (USD 3042).	CPA is cheapest with obvious effect in China, while leflunomide is a cost-effective alternative therapy. CLB, RTX, CsA, and TAC are higher cost and lower utility than expected.

Abbreviations: SDNS, steroid-dependent nephrotic syndrome; SRNS, steroid-resistant nephrotic syndrome; CNI, calcineurin inhibitor; CER, cost-effectiveness ratio.

## Data Availability

Not applicable.

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
