# Peer review of "Immunosuppressive Agent Options for Primary Nephrotic Syndrome: A Review of Network Meta-Analyses and Cost-Effectiveness Analysis"

_medicina, 2023, doi:10.3390/medicina59030601_

Round 1

Reviewer 1 Report

The review "Treatment of Refractory Glomerular Diseases: Challenges and Solutions "is interesting and well written. I suggest in the end of conclusion to add some data about "new" no immunosoppressive agents to treat glomerulonephritis such as SGLT 2 inhibitors

Author Response

Comment:

I suggest in the end of conclusion to add some data about "new" no immunosuppressive agents to treat glomerulonephritis such as SGLT 2 inhibitors.

Thank you for your thoughtful suggestion. I have decided that it would be difficult to incorporate non-immunosuppressive agents, such as SGLT2i, into the context of this review, since it is quite far from the scope of the review. I would like to leave the concluding statement as is.

Reviewer 2 Report

1,Line 98 : please rephrase , It is not recommended to use the first person singular (Ι ..) in writing scientific articles, please modify the text.

2. The effect of long-term treatment with corticosteroids in children has been derived in large studies and not only in nephrotic syndrome but in a variety of diseases. Therefore, at least in pediatrics, it is imperative to use other therapeutic agents. This should be sufficiently emphasized.

3.In pediatric nephrology, the initial treatment of nephrotic syndrome is prednisolone based on the guidelines. However, it should be emphasized that when there is no response to corticosteroids then a kidney biopsy is performed before adding alternative immunosuppressive agent. Rituximub is not given as a first-line drug and is always given after biopsy in patients with resistast nephrotic syndrome  who do not respond to second-line therapy such as cyclosporine etc.

3.Overall, the review on immunosuppressive agents used in nephrotic syndrome is incomplete. There are clear guidelines. Some agents are no longer recommended. Moreover, despite the new method used in this work, there is no novelty.

Thank you

Author Response

1. Line 98 : please rephrase , It is not recommended to use the first person singular (Ι ..) in writing scientific articles, please modify the text.

Thank you for your comments on the use of the subject in academic papers. I would like to rephrase them properly in the revised version (Line 98, 184, 187 and 197).

2. The effect of long-term treatment with corticosteroids in children has been derived in large studies and not only in nephrotic syndrome but in a variety of diseases. Therefore, at least in pediatrics, it is imperative to use other therapeutic agents. This should be sufficiently emphasized.

Thank you for your thoughtful suggestion. I would like to emphasize the importance of immunosuppression to reduce the use of steroids, not only for nephrotic syndrome but also for many other diseases, because of the need for long-term administration of steroids (Line 113-115).

3. In pediatric nephrology, the initial treatment of nephrotic syndrome is prednisolone based on the guidelines. However, it should be emphasized that when there is no response to corticosteroids then a kidney biopsy is performed before adding alternative immunosuppressive agent. Rituximub is not given as a first-line drug and is always given after biopsy in patients with resistant nephrotic syndrome who do not respond to second-line therapy such as cyclosporine etc.

Thank you for sharing thoughts on treatment for nephrotic syndrome in pediatric nephrology. I have included the indication of renal biopsy and the current treatment algorithm in the second section (Current Applications of Immunosuppressive Agents for Primary NS Listed in Guide-lines, Line 139-141 and 152-154)

4. Overall, the review on immunosuppressive agents used in nephrotic syndrome is incomplete. There are clear guidelines. Some agents are no longer recommended. Moreover, despite the new method used in this work, there is no novelty.

Thank you for your comments on the concept. This review article has been prepared with the intent not to blindly follow guidelines without taking into account regional differences in economic status and drug availability. I believe that this review is necessary because it is not always sufficient in the guideline to focus on network meta-analyses and cost-effectiveness studies to discuss them properly.

Round 2

Reviewer 2 Report

Dear authos ,  I think that your paper can be accepted in present form.

Thank you